# KANFIS: A Neuro-Symbolic Framework for Interpretable and Uncertainty-Aware Learning

**Binbin Yong**[1]  **Haoran Pei**[1]  **Jun Shen**[2]  **Haoran Li**[3]  **Qingguo Zhou**[1]  **Zhao Su**[1]

## Abstract

Adaptive Neuro-Fuzzy Inference System (ANFIS) was designed to combine the learning capabilities of neural network with the reasoning transparency of fuzzy logic. However, conventional ANFIS architectures suffer from structural complexity, where the product-based inference mechanism causes an exponential explosion of rules in high-dimensional spaces. We herein propose the **K**olmogorov-**A**rnold **N**euro-**F**uzzy **I**nference **S**ystem (KANFIS), a compact neuro-symbolic architecture that unifies fuzzy reasoning with additive function decomposition. KANFIS employs an additive aggregation mechanism, under which both model parameters and rule complexity scale linearly with input dimensionality rather than exponentially. Furthermore, KANFIS is compatible with both Type-1 (T1) and Interval Type-2 (IT2) fuzzy logic systems, enabling explicit modeling of uncertainty and ambiguity in fuzzy representations. By using sparse masking mechanisms, KANFIS generates compact and structured rule sets, resulting in an intrinsically interpretable model with clear rule semantics and transparent inference processes. Empirical results demonstrate that KANFIS achieves competitive performance against representative neural and neuro-fuzzy baselines.

## 1. Introduction

Interpretable machine learning has re-emerged as a central research focus as modern neural networks continue to grow in scale, complexity, and opacity, raising increasing

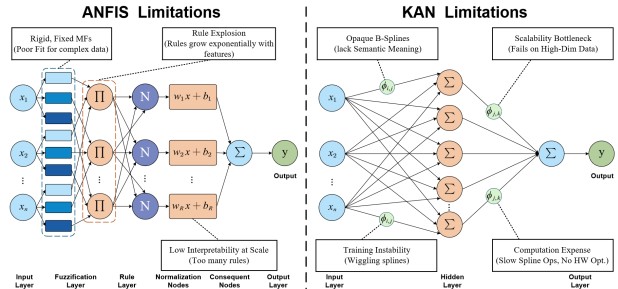

*Figure 1.* **Limitations of Existing Interpretable Models.** Conventional ANFIS and KAN models suffer from several limitations in terms of interpretability and model complexity.

concerns about transparency, reliability, and trustworthiness (Somvanshi et al., 2025; Minh et al., 2022). Despite their impressive empirical performance across domains such as vision, language, and time-series forecasting, conventional architectures such as Multilayer Perceptron (MLP) and Transformer often operate as black boxes (Ibrahim & Shafiq, 2023; Hassija et al., 2024), offering limited insights into their internal reasoning processes or decision structures. This lack of transparency becomes particularly critical in high-stake applications (Liao et al., 2024; Khan et al., 2024), including energy forecasting, finance, and autonomous systems, where model reliability, uncertainty awareness, and interpretability are increasingly regarded as top priority requirements (Arrieta et al., 2020; Rudin et al., 2022).

Consequently, there has been growing interest in developing models that combine strong predictive performance with intrinsic interpretability. One representative direction is the Neuro-Fuzzy System (NFS), which bridges statistical learning with symbolic reasoning (Minh et al., 2022; Hassija et al., 2024). The Adaptive Neuro-Fuzzy Inference System (ANFIS) represents a representative NFS architecture that combines fuzzy rule-based reasoning with gradient-based parameter learning, enabling models to express structured linguistic rules while adapting to data. However, these traditional frameworks typically rely on fixed linear consequent mappings and static rule structures, which severely limit their expressivity and scalability, particularly in high-dimensional or highly nonlinear settings. Subsequent developments have sought to mitigate these limitations by introducing self-organizing rule structures that evolve from

---

[1]School of Information Science and Engineering, Lanzhou University, China [2]School of Computing and Information Technology, University of Wollongong, Australia [3]Department of Data Science and Artificial Intelligence, Monash University, Australia. Correspondence to: Qingguo Zhou <zhouqg@lzu.edu.cn>, Zhao Su <szhao2024@lzu.edu.cn>.

*Proceedings of the $43^{rd}$ International Conference on Machine Learning*, Seoul, South Korea. PMLR 306, 2026. Copyright 2026 by the author(s).

data streams (Meng et al., 2025), Interval Type-2 (IT2) fuzzy sets for enhanced uncertainty modeling (Yao et al., 2025), and hybrid neuro-fuzzy architectures designed to capture hierarchical or complex features (Shihabudheen & Pillai, 2018). Despite these advancements, a fundamental trade-off persists: most neuro-fuzzy systems struggle to simultaneously achieve strong nonlinear approximation, compact and interpretable rule structures, and robustness under uncertainty, particularly as model complexity increases.

In parallel, recent work on the Kolmogorov-Arnold Network (KAN) (Somvanshi et al., 2025) demonstrates the expressive power of functional composition as a replacement for conventional linear weight matrices. By learning adaptive univariate basis functions along each path of a computational graph, KAN significantly enhances representational capacity while maintaining a structured decomposition aligned with the classical Kolmogorov-Arnold superposition theorem. Although KAN does not provide explicitly interpretable rules, these developments still illustrate the potential advantages of replacing fixed linear transformations with learnable nonlinear operators, particularly in systems that inherently rely on compositional mappings, such as fuzzy inference models. Recent systematic reviews on fuzzy systems and interpretable machine learning highlight that evolving and hybrid fuzzy architectures have been increasingly studied for their ability to balance approximation power and rule interpretability, yet they still face challenges in scalability and structural transparency (Gu et al., 2023; Ouifak & Idri, 2025). As illustrated in Figure 1, existing interpretable models exhibit complementary limitations: ANFIS suffers from rigid structures and rule explosion, whereas KAN encounters semantic opacity and scalability bottlenecks in high-dimensional regimes.

Motivated by the aforementioned problems, we propose the **K**olmogorov-**A**rnold **N**euro-**F**uzzy **I**nference **S**ystem (KANFIS). This architecture integrates the symbolic reasoning of ANFIS with the additivity theorem of Kolmogorov-Arnold networks. Specifically, we design the network edges to function as learnable membership functions, which are then aggregated using the additivity principle. This configuration enhances expressive power without sacrificing interpretability, as it naturally accommodates Type-1 (T1) and IT2 reasoning to explicitly encode uncertainty, while the functional operators provide transparent pathways describing how each input feature contributes to the prediction. Furthermore, to guarantee model compactness and interpretability, we further introduce specific regularization terms. One term induces sparsity to control the number of active features in each rule, while the other enforces diversity to maintain distinctiveness between rules. Consequently, KANFIS achieves competitive prediction accuracy while generating concise, human-readable rule-based explanations.

Our contributions are summarized as follows:

- We propose KANFIS, the first framework to realize the symbolic reasoning of ANFIS within a Kolmogorov-Arnold-style additive architecture. By leveraging the superposition of learnable functions, KANFIS achieves exceptional nonlinear approximation capabilities.

- We achieve a compact architectural design by leveraging the Kolmogorov-Arnold additivity principle together with a dedicated regularization strategy, enabling efficient function approximation with fewer rules while suppressing redundant features and rule proliferation.

- Benefiting from its unique design, KANFIS maintains compact and well-structured rule representations in high-dimensional settings, mitigating rule explosion and interpretability degradation observed in conventional ANFIS.

## 2. Related Work

Our work focuses on intrinsically interpretable learning models with transparent reasoning processes. A broad range of related studies has explored interpretable architectures (Tursunalieva et al., 2024), neuro-fuzzy inference systems, and function-based model representations (Somvanshi et al., 2025), forming the foundation of this line of research.

### 2.1. Intrinsic Interpretability and Additive Models

The notion of interpretability is inherently ambiguous and has been shown to encompass multiple, often conflated, concepts (Lipton, 2018). In the context of Explainable AI (XAI) (Gao et al., 2024; Mehdiyev et al., 2025), a key distinction lies between post-hoc explanations and intrinsically interpretable models. Post-hoc methods such as SHAP (Lundberg & Lee, 2017; Enouen & Liu, 2025) and LIME (Ribeiro et al., 2016) provide local approximations of black-box predictors, yet their faithfulness to the underlying decision process is frequently questioned. Rudin (Rudin, 2019; Rudin et al., 2022) argues that reliable transparency should instead be achieved through models that are interpretable by design. Generalized Additive Models (GAMs) constitute a foundational class of intrinsically interpretable models, expressing predictions as sums of feature-wise effects. Recent variants, such as M-GAM, extend GAMs to practical settings by explicitly handling missing data while preserving interpretability (McTavish et al., 2024). Neural Additive Models (NAMs) (Agarwal et al., 2021) enhance GAMs by parameterizing univariate functions with neural networks, substantially improving expressive power while retaining additive transparency. By enabling direct inspection of learned feature functions, NAMs offer an effective

balance between accuracy and interpretability, a property further examined by Mariotti et al. (Mariotti et al., 2023). To capture feature interactions, several extensions have been proposed, including GA$^2$M with pairwise terms (Lou et al., 2013), NODE-GAM with tree-based components (Chang et al., 2021), and GAMI-Net with hierarchical regularization (Yang et al., 2021). Nevertheless, modeling complex or high-order interactions in a globally interpretable manner remains challenging. In contrast, our approach exploits the compositional, rule-based structure of fuzzy systems to encode rich interactions while maintaining semantic transparency.

## 2.2. Adaptive Neuro-Fuzzy Inference System (ANFIS)

ANFIS combine the learning capabilities of neural networks with the transparent, rule-based reasoning of fuzzy logic (Rawal et al., 2025). The seminal ANFIS architecture maps a Takagi-Sugeno fuzzy inference system into a five-layer adaptive network, enabling interpretable "IF-THEN" rules alongside gradient-based optimization. ANFIS has been widely applied in control, regression, and pattern recognition tasks due to this balance of adaptability and transparency. However, as input dimensionality grows, the number of fuzzy rules increases exponentially, leading to the curse of dimensionality and limiting scalability in high-dimensional or industrial data scenarios (Xue et al., 2023; Cui et al., 2022). To address these limitations, a wide range of extensions have been proposed. Clustering-based rule generation and optimization strategies reduce rule complexity, while recurrent and deep fuzzy architectures enhance modeling capacity for temporal and high-dimensional data. For example, fuzzy recurrent stochastic configuration networks integrate TSK fuzzy inference with recurrent stochastic mechanisms, achieving robust performance on industrial datasets (Wang & Dang, 2025). Deep spatio-temporal fuzzy models have been applied to geospatial forecasting tasks such as NDVI prediction, effectively capturing spatio-temporal correlations while maintaining interpretability (Su et al., 2025a). Hybrid structure-learnable models like SL-ANFIS-LSTM combine fuzzy reasoning with LSTM dynamics for ultra-short-term photovoltaic power forecasting, balancing nonlinear temporal modeling and transparent rule extraction (Su et al., 2025b). Recent surveys further highlight the continuous evolution of neuro-fuzzy architectures toward scalable, high-dimensional, and interpretable designs (Gu et al., 2023; Apiecionek, 2025). Despite these advances, most contemporary neuro-fuzzy systems still rely on conventional neural parameterizations in antecedent or consequent parts, limiting parameter efficiency and expressive power in very high-dimensional tasks. In contrast, our approach introduces KAN as a parameter-efficient, compositional alternative within the fuzzy inference framework, preserving semantic transparency while encoding complex interactions

naturally. This design addresses the longstanding trade-off between interpretability and expressive capacity, situating our work at the intersection of structured fuzzy reasoning and function-based neural architectures.

## 2.3. Kolmogorov-Arnold Network (KAN)

The Kolmogorov-Arnold representation theorem states that any multivariate continuous function can be represented as a finite composition of univariate functions and addition, offering a theoretical foundation for structured function approximation. Historically, this theorem was largely dismissed in neural network design due to smoothness and constructive concerns. Recently, Liu et al. (Liu et al., 2024) revitalized this theory through KAN, which replaces conventional linear weight matrices with learnable univariate functions along edges, effectively embedding functional decomposition into the network architecture. Unlike MLP with fixed activation functions, KAN allow each edge to have a learnable activation function, often parameterized as B-splines. This design improves both expressive efficiency and interpretability, as the learned univariate functions can be visualized and semantically analyzed (Somvanshi et al., 2025; Ta, 2025). Extensions of KAN explore alternative bases, such as sinusoidal activations (SineKAN) (Reinhardt et al., 2025) or hybrid B-spline + radial basis functions (BSRBF-KAN) (Ta, 2025), which improve approximation fidelity and training stability across various tasks. Empirical studies demonstrate that KAN can match or outperform larger MLP in both data fitting and symbolic regression tasks while maintaining significantly fewer parameters (Liu et al., 2024; Wang et al., 2025). Complementary theoretical work establishes generalization bounds for KAN, showing that the structured composition of univariate functions allows for a rigorous analysis of the complexity of the model (Zhang & Zhou, 2024). In this work, we reinterpret these learnable univariate functions as adaptive fuzzy membership functions, grounding KAN within a semantic logical framework. This enables a natural integration with fuzzy inference systems, achieving compositional transparency and interpretable reasoning in high-dimensional predictive modeling.

## 2.4. Theoretical Connection: Splines and Fuzzy Sets

The integration of KAN into fuzzy inference systems is theoretically supported by the well-established connection between B-splines and fuzzy logic. Classical results show that the B-spline neural network can be interpreted as a class of fuzzy systems under specific overlap and normalization conditions . This equivalence provides a rigorous mathematical foundation for viewing learned univariate functions as fuzzy membership functions, enabling a principled fusion of functional approximation and symbolic reasoning. Traditional B-spline networks, however, typically relied on

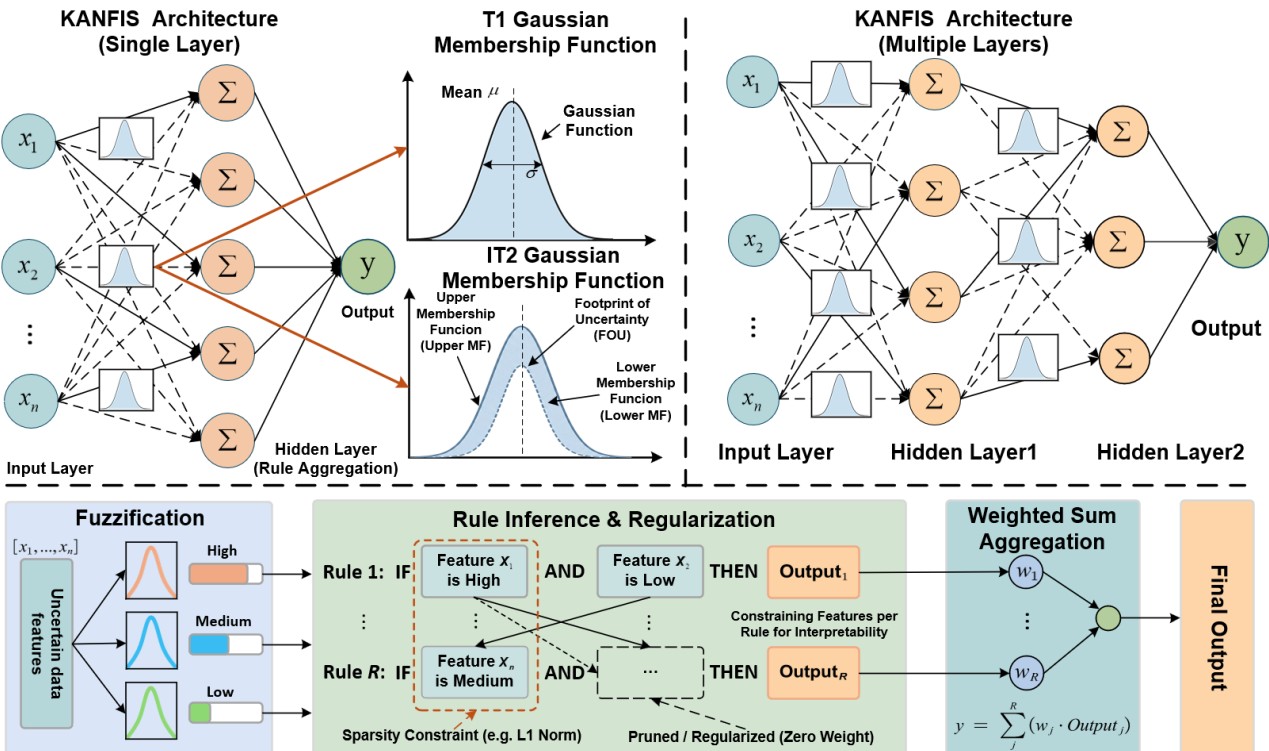

*Figure 2.* **Top:** KANFIS structures with one and two layers, respectively. Dashed lines indicate paths that may be selected during learning. **Bottom:** Computational flow of the model. $x$ denotes the input features, $y$ the predicted output, and $\omega$ the weight of each rule, reflecting the influence of each rule pattern on the final prediction.

fixed grid structures, constraining their adaptability and limiting expressive capacity in high-dimensional tasks. KAN introduces adaptive, learnable grids along edges, allowing each univariate function to self-organize in response to data. By incorporating these adaptive B-spline functions into the antecedent part of fuzzy rules, our approach bridges modern spline-based networks with interpretable logical systems, yielding models that are both mathematically sound and semantically meaningful. Recent work has further extended this connection, demonstrating that structured spline expansions within KAN can rigorously approximate continuous multivariate functions while preserving interpretability (Ta, 2025; Reinhardt et al., 2025; Zhang & Zhou, 2024). These advances provide strong theoretical justification for our KANFIS framework, where fuzzy membership semantics are naturally encoded in learnable univariate components of KAN.

## 3. Methodology

The KANFIS model reorganizes the classical ANFIS architecture based on the Kolmogorov-Arnold representation theorem, resulting in a simple yet expressive interpretable neural network. The overall architecture and computational flow of KANFIS are illustrated in Figure 2. This section presents a detailed description of the model architecture and

the proposed methodology.

### 3.1. Learnable Membership Functions

Traditional neural networks rely on crisp scalar weights $w_{i,j}$ to modulate the contribution of input $x_i$ to hidden neuron $j$. In contrast, KANFIS replaces each such weight with a set of $K$ learnable fuzzy functional mappings. To accommodate diverse data distributions and modeling requirements, the proposed architecture supports both T1 and IT2 configurations. In the T1 setting, we incorporate three distinct membership functions to capture local nonlinearities: Gaussian, Generalized Bell, and Sigmoid. Detailed formulations for the Bell and Sigmoid functions are provided in **Appendix C** for completeness. Taking the Gaussian function as the primary example, the membership grade for an input $x_i$ is parameterized by a learnable center $\mu_{i,j,k}$ and width $\sigma_{i,j,k}$:

$$\mathcal{M}_{i,j,k}(x_i) = \exp\left(-\frac{(x_i - \mu_{i,j,k})^2}{2\sigma_{i,j,k}^2}\right). \qquad (1)$$

To enable the model to capture aleatoric uncertainty alongside nonlinearity, we extend the Gaussian basis to the IT2 setting. Each basis function is parameterized by a center $\mu_{i,j,k}$ and a pair of standard deviations $(\sigma_{i,j,k}^{(l)}, \sigma_{i,j,k}^{(u)})$ that

*Table 1.* **Comparative experimental results.** We adopt a clustering-based IT2-ANFIS to ensure its suitability as a baseline under high-dimensional feature settings. The symbol '-' indicates that the model cannot be trained due to the curse of dimensionality.

| Dataset | Metrics | IT2-KANFIS | T1-KANFIS | MLP | T1-ANFIS | IT2-ANFIS | KAN |
|---|---|---|---|---|---|---|---|
| CCPP | MAPE | 0.7047 | 0.6777 | 0.7164 | 0.6759 | 0.6696 | **0.6389** |
| | RMSE | 4.1240 | 3.9542 | 4.1883 | 3.9980 | 4.0047 | **3.9313** |
| | MAE | 3.1975 | 3.0760 | 3.2553 | 3.0688 | 3.0439 | **2.8988** |
| Parkinsons | MAPE | 14.891 | **14.477** | 19.604 | 16.167 | 15.446 | 14.610 |
| | RMSE | **0.0397** | 0.0405 | 0.0527 | 0.0449 | 0.0433 | 0.0404 |
| | MAE | 0.0291 | **0.0289** | 0.0389 | 0.0320 | 0.0310 | 0.0293 |
| BCW | Acc | **0.9912** | 0.9912 | 0.9912 | – | 0.9737 | 0.9737 |
| | F1 | **0.9912** | 0.9912 | 0.9912 | – | 0.9737 | 0.9736 |
| | AUROC | **0.9990** | 0.9987 | 0.9845 | – | 0.9954 | 0.9940 |
| Spambase | Acc | **0.9392** | 0.9381 | 0.9175 | – | 0.8827 | – |
| | F1 | **0.9392** | 0.9379 | 0.9172 | – | 0.8821 | – |
| | AUROC | **0.9797** | 0.9769 | 0.9599 | – | 0.9397 | – |
| MHR | Acc | **0.7931** | 0.7931 | 0.7094 | 0.7438 | 0.6847 | 0.7783 |
| | F1 | **0.7942** | 0.7941 | 0.7028 | 0.7428 | 0.6714 | 0.7777 |
| | AUROC | **0.9057** | 0.9157 | 0.8427 | 0.8881 | 0.8535 | 0.9071 |

satisfy

$$\sigma_{i,j,k}^{(l)} > 0, \qquad \sigma_{i,j,k}^{(u)} > \sigma_{i,j,k}^{(l)}. \qquad (2)$$

Given an input value $x_i$, its upper and lower membership degrees are

$$\mathrm{UMF}_{i,j,k}(x_i) = \exp\left(-\frac{(x_i - \mu_{i,j,k})^2}{2\left(\sigma_{i,j,k}^{(u)}\right)^2}\right), \qquad (3)$$

$$\mathrm{LMF}_{i,j,k}(x_i) = \exp\left(-\frac{(x_i - \mu_{i,j,k})^2}{2\left(\sigma_{i,j,k}^{(l)}\right)^2}\right). \qquad (4)$$

These upper and lower surfaces define a Footprint of Uncertainty (FOU) associated with the connection $i \rightarrow j$. To obtain a crisp quantity during inference, we employ the center-of-sets type-reduction:

$$\phi_{i,j,k}(x_i) = \frac{\mathrm{UMF}_{i,j,k}(x_i) + \mathrm{LMF}_{i,j,k}(x_i)}{2}. \qquad (5)$$

The quantity $\phi_{i,j,k}(x_i)$ measures how strongly input $x_i$ activates the $k$-th fuzzy basis along the $(i, j)$ edge. We provide a theoretical proof justifying the validity of this additive aggregation scheme in **Appendix B**.

### 3.2. Functional Edges as Fuzzy Rule Antecedents

Rather than using a single weight, each edge aggregates $K$ basis memberships:

$$e_{i,j}(x_i) = \sum_{k=1}^{K} \phi_{i,j,k}(x_i). \qquad (6)$$

This quantity plays the role of a *soft rule antecedent*. Specifically, the interaction between input $x_i$ and hidden unit $j$ is interpreted as:

*"If feature $x_i$ belongs to one of the fuzzy sets associated with unit $j$, then the rule fires with strength $e_{i,j}(x_i)$."*

Because different features activate different fuzzy sets, the model naturally learns localized nonlinear transformations without requiring explicit rule enumeration.

### 3.3. Kolmogorov-Arnold Aggregation Over Edges

For hidden unit $j$, all incoming fuzzy edges are combined through the Kolmogorov-style summation:

$$h_j = \sum_{i=1}^{d_{\mathrm{in}}} e_{i,j}(x_i). \qquad (7)$$

Equation (7) is motivated directly by the Kolmogorov-Arnold representation theorem, which states that any continuous multivariate function can be represented as a finite superposition of univariate functions and addition. In KAN-FIS, each $\phi_{i,j,k}$ is a univariate fuzzy transformation, and $e_{i,j}$ aggregates these univariate transforms, and $h_j$ forms a superposition across input dimensions. The resulting hidden representation thus acts as a learned nonlinear embedding constructed entirely through fuzzy functional components.

### 3.4. Deep Stacking of Fuzzy-Functional Layers

Multiple KANFIS layers can be stacked to form a deep fuzzy inference hierarchy. Let $\mathbf{h}^{(0)} = \mathbf{x}$ denote the input.

*Table 2.* **Interpretability analysis of fuzzy rules on the CCPP dataset.** The six most important rules learned by the model and their associated physical principles are presented in the table, with their validity supported by evidence from the cited relevant physical studies.

| IF | THEN | Interpretation |
|---|---|---|
| AT is HIGH | $-0.3407 \cdot (7.26 \cdot \mathcal{M}_{AT}(x_{AT}))$ | High ambient temperature reduces air density and mass flow, lowering power output (Saravanamuttoo et al., 2001). |
| V is LOW & AP is MED & RH is HIGH | $0.3070 \cdot (2.96 \cdot \mathcal{M}_V(x_V) + 0.14 \cdot \mathcal{M}_{AP}(x_{AP}) + 0.16 \cdot \mathcal{M}_{RH}(x_{RH}))$ | Low exhaust vacuum and high relative humidity reduce back pressure and increase specific heat, creating optimal operating conditions (Cengel & Boles, 2002). |
| AT is MED & AP is HIGH | $0.2541 \cdot (3.38 \cdot \mathcal{M}_{AT}(x_{AT}) + 0.48 \cdot \mathcal{M}_{AP}(x_{AP}))$ | Higher ambient pressure increases air density, enhancing turbine intake flow and efficiency even at moderate temperatures (Saravanamuttoo et al., 2001). |
| V is LOW & RH is LOW | $0.2038 \cdot (2.31 \cdot \mathcal{M}_V(x_V) + 1.61 \cdot \mathcal{M}_{RH}(x_{RH}))$ | Low exhaust vacuum dominates efficiency; reduced condenser back pressure ensures high power output even at low humidity (Moran et al., 2010). |
| AT is MED & V is HIGH | $-0.1831 \cdot (3 \cdot \mathcal{M}_{AT}(x_{AT}) + 0.53 \cdot \mathcal{M}_V(x_V))$ | High exhaust vacuum increases condenser back pressure, restricting steam expansion and offsetting medium-temperature benefits (Moran et al., 2010). |
| AT is HIGH & RH is LOW | $-0.1351 \cdot (3.88 \cdot \mathcal{M}_{AT}(x_{AT}) + 0.93 \cdot \mathcal{M}_{RH}(x_{RH}))$ | Low relative humidity implies lower specific heat, slightly reducing gas turbine output at similar temperatures (Cengel & Boles, 2002). |

For the $\ell$-th fuzzy layer, the transformation is:

$$\mathbf{h}^{(\ell)} = \mathcal{F}^{(\ell)}\left(\mathbf{h}^{(\ell-1)}\right), \tag{8}$$

where $\mathcal{F}^{(\ell)}$ encapsulates Equation (3)-(7) applied to all units in the layer. To stabilize training and maintain numeric consistency across layers, a normalization is applied after each layer except the final one:

$$\mathbf{h}^{(\ell)} \leftarrow \mathrm{Norm}(\mathbf{h}^{(\ell)}), \qquad \ell < L. \tag{9}$$

This structure resembles a deep neural network, but with crisp weights replaced entirely by IT2 fuzzy functional edges, resulting in richer representational capacity and improved robustness to uncertainty.

### 3.5. Defuzzification via Linear Projection

After the final fuzzy layer, its output $\mathbf{h}^{(L)}$ is mapped to the prediction space via a linear defuzzification layer:

$$\hat{\mathbf{y}} = W\mathbf{h}^{(L)} + b. \tag{10}$$

This operation aggregates all activated fuzzy rules into a single crisp prediction. It can be interpreted as a generalized consequent layer of a Takagi-Sugeno fuzzy system, where the antecedent firing strengths are given by the KANFIS pipeline and the consequent functions are linear mappings.

### 3.6. Regularization

While KANFIS possesses strong approximation capabilities, unconstrained training of fuzzy systems can lead to redundant rules with dense antecedents, which severely hinders interpretability. To ensure the learned rules are both concise

(sparse) and diverse (distinct), we augment the primary task loss $\mathcal{L}_{\mathrm{task}}$ with two specific regularization objectives. The total objective function is formulated as:

$$\mathcal{L}_{\mathrm{total}} = \mathcal{L}_{\mathrm{task}} + \lambda_s \mathcal{R}_{\mathrm{sparse}} + \lambda_d \mathcal{R}_{\mathrm{distinct}}, \tag{11}$$

where $\lambda_s$ and $\lambda_d$ are hyperparameters that control the strength of the regularization.

**Entropy-based Antecedent Sparsity.** To improve human readability, each fuzzy rule should ideally condition on a minimal subset of relevant input features (typically 2-3 antecedents). We introduce a learnable soft-masking matrix $\mathbf{M} \in [0,1]^{D \times K}$, where $M_{i,k}$ represents the importance of input feature $i$ to rule $k$. To avoid the optimization difficulties of discrete selection, we relax the problem by minimizing the pixel-wise binary entropy of $\mathbf{M}$. This forces the mask values to converge towards either 0 (irrelevant) or 1 (selected):

$$\mathcal{R}_{\mathrm{sparse}} = \frac{1}{D \cdot K} \sum_{k=1}^{K} \sum_{i=1}^{D} \Phi(M_{i,k}), \tag{12}$$

where $\Phi(p) = -p \log p - (1-p) \log(1-p)$. By minimizing $\mathcal{R}_{\mathrm{sparse}}$, the model is encouraged to prune unnecessary connections, naturally revealing the underlying sparse structure of the data.

**Rule Distinctiveness.** A common pathology in neuro-fuzzy learning is mode collapse, where multiple rules converge to characterize identical local regions, leading to redundancy. To enforce diversity, we propose a distinctiveness regularization term that penalizes the overlap between the firing patterns of different rules. Let $\mathbf{w}_j \in \mathbb{R}^B$ denote the

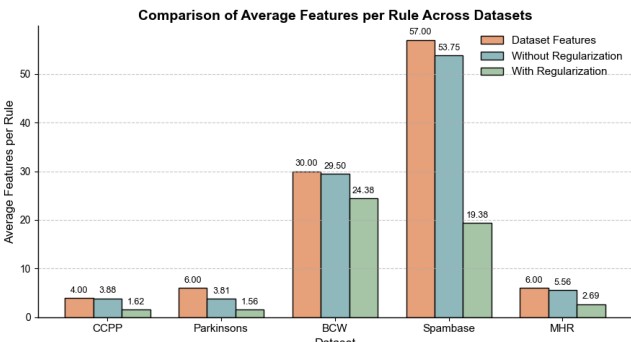

*Figure 3.* **Comparison of the average number of features per rule across datasets.** The figure illustrates the comparison between the number of features used in the rules and the original number of features in each dataset, depending on whether regularization is applied.

vector of firing strengths for rule $j$ across a mini-batch of size $B$. We minimize the pairwise cosine similarity between all distinct pairs of rules:

$$\mathcal{R}_{\text{distinct}} = \sum_{j=1}^{K} \sum_{k=j+1}^{K} \frac{\mathbf{w}_j^\top \mathbf{w}_k}{\|\mathbf{w}_j\|_2 \|\mathbf{w}_k\|_2}. \quad (13)$$

Minimizing this term encourages the rules to be orthogonal in their activation space, ensuring that each rule specializes in a unique sub-region of the input domain, thereby maximizing the informational capacity of the limited rule base.

### 3.7. Overall Forward Computation

The complete forward propagation of the KANFIS architecture can be summarized as

$$\mathbf{h}^{(0)} = \mathbf{x}, \quad (14)$$

$$\mathbf{h}^{(\ell)} = \mathcal{F}^{(\ell)}\left(\mathbf{h}^{(\ell-1)}\right), \qquad \ell = 1, \dots, L, \quad (15)$$

$$\hat{\mathbf{y}} = W\mathbf{h}^{(L)} + b. \quad (16)$$

Together, Equation (2)-(16) constitute a complete mathematical description of the proposed KANFIS model. Regarding the universal approximation capability of KANFIS, we provide the proof in the **Appendix A**.

## 4. Experiments

To evaluate the capabilities of the proposed model, we select datasets spanning diverse domains and conduct comparative experiments against baseline models, including MLP, ANFIS and KAN. These datasets exhibit significant variations in terms of sample size and feature dimensionality, providing a challenging testbed for performance evaluation.

### 4.1. Comparison Experiments

Table 1 presents the comparative experimental results of KANFIS and other baseline models. The results demonstrate that KANFIS consistently achieves superior performance across multiple datasets, outperforming all competing methods. To further strengthen the credibility of the performance evaluation, **Appendix D** additionally includes experiments on several higher-dimensional and larger-scale datasets, as well as comparisons with other interpretable models that are relatively less correlated with the proposed framework. Moreover, unlike MLPs, which typically rely on deep architectures to capture complex patterns, KANFIS often attains optimal convergence with only a single-layer structure, highlighting its structural efficiency.

Traditional ANFIS methods are not well-suited for high-dimensional datasets. To enable a comprehensive comparison, T1-ANFIS retains the conventional partitioning strategy, while IT2-ANFIS adopts scatter partitioning, allowing meaningful evaluation in high-dimensional settings. A closer inspection of the comparative results reveals significant limitations in existing approaches that our model successfully overcomes: While traditional ANFIS performs adequately in low-dimensional spaces, it suffers severely from the curse of dimensionality. In high-dimensional settings, standard grid partitioning becomes computationally intractable due to the exponential explosion of rules. Although alternative techniques like scatter partitioning (employed in IT2-ANFIS) allow for execution on high-dimensional data, our results show that this adaptation comes at the cost of a significant drop in accuracy and a degradation of the model's inherent interpretability. The standard KAN architecture exhibits poor scalability. We observed that the computational cost of KAN escalates drastically with the increase in data volume and feature dimensionality. Consequently, on larger datasets such as Spambase and ForestCover, the baseline KAN failed to complete the training process due to prohibitive computational demands. Overall, KANFIS stands out as the most robust interpretable learner, delivering high performance without sacrificing model transparency. We explicitly demonstrate this interpretability through a case study in the subsequent section.

### 4.2. Interpretability Analysis

In this section, we demonstrate the interpretability of our model using an industrial regression dataset, the Combined Cycle Power Plant (CCPP), and a medical classification dataset, the Medical Health Records (MHR). CCPP includes four features, i.e., Ambient Temperature (AT), Exhaust Vacuum (V), Ambient Pressure (AP), and Relative Humidity (RH), with the target being Energy Output. MHR contains seven features, i.e., Age, Systolic Blood Pressure (Sys-

*Table 3.* **Interpretability analysis of fuzzy rules on the MHR dataset.** The table presents the six most important rules learned by the model along with their corresponding medical principles, and their validity is supported by evidence from the cited relevant studies. For clarity, only the class formula that best matches each rule is reported in the THEN clause.

| IF | THEN | Interpretation |
|---|---|---|
| SystolicBP is HIGH & DiastolicBP is MED | $P_{highrisk} = 4.0611 \cdot (7.28 \cdot \mathcal{M}_{\text{SystolicBP}}(x_{\text{SystolicBP}}) + 2.09 \cdot \mathcal{M}_{\text{DiastolicBP}}(x_{\text{DiastolicBP}}))$ | Elevated systolic and diastolic BP increase cardiovascular risk (He & Whelton, 1999). |
| Age is LOW & SystolicBP is HIGH & BS is LOW | $P_{lowrisk} = 0.814 \cdot (3.59 \cdot \mathcal{M}_{\text{Age}}(x_{\text{Age}}) + 0.94 \cdot \mathcal{M}_{\text{SystolicBP}}(x_{\text{SystolicBP}}) + 1.56 \cdot \mathcal{M}_{\text{BS}}(x_{\text{BS}}))$ | Young age with slightly high BP and low glucose indicates low overall risk (D'Agostino et al., 2008). |
| DiastolicBP is LOW | $P_{highrisk} = 0.2811 \cdot (4.49 \cdot \mathcal{M}_{\text{DiastolicBP}}(x_{\text{DiastolicBP}}))$ | Low diastolic BP may reduce perfusion and raise cardiovascular risk (McEvoy et al., 2016). |
| SystolicBP is HIGH & BS is LOW | $P_{lowrisk} = 1.4001 \cdot (1.48 \cdot \mathcal{M}_{\text{SystolicBP}}(x_{\text{SystolicBP}}) + 3.81 \cdot \mathcal{M}_{\text{BS}}(x_{\text{BS}}))$ $P_{mediumrisk} = 1.1433 \cdot (1.48 \cdot \mathcal{M}_{\text{SystolicBP}}(x_{\text{SystolicBP}}) + 3.81 \cdot \mathcal{M}_{\text{BS}}(x_{\text{BS}}))$ | Slightly high SBP with low glucose corresponds to low-to-moderate risk (D'Agostino et al., 2008). |
| SystolicBP is MED & BS is LOW & BodyTemp is LOW | $P_{mediumrisk} = 1.0646 \cdot (2.2 \cdot \mathcal{M}_{\text{SystolicBP}}(x_{\text{SystolicBP}}) + 3.21 \cdot \mathcal{M}_{\text{BS}}(x_{\text{BS}}) + 1.34 \cdot \mathcal{M}_{\text{BodyTemp}}(x_{\text{BodyTemp}}))$ | High SBP with low glucose and normal/low temp indicates moderate risk (D'Agostino et al., 2008). |
| Age is HIGH | $P_{highrisk} = -0.3013 \cdot (2.81 \cdot \mathcal{M}_{\text{Age}}(x_{\text{Age}}))$ | Advanced age independently increases cardiovascular risk (D'Agostino et al., 2008). |

tolicBP), Diastolic Blood Pressure (DiastolicBP), Blood Sugar (BS), Body Temperature (BodyTemp), and Heart Rate (HeartRate), with the target being Risk Level. The features and targets of both datasets are supported by domain-specific prior knowledge, providing a reference to verify whether the model captures genuine physical or physiological patterns rather than spurious correlations. Furthermore, to better highlight interpretability, we reduced the number of rules in the model. KANFIS possesses structural transparency, allowing symbolic IF-THEN rules to be explicitly extracted. Table 2 presents a subset of the dominant rules extracted from the model trained on the CCPP dataset, while Table 3 shows a subset of the dominant rules extracted from the model trained on the MHR dataset. As shown in the two tables above, and supported by references from relevant domain literature, we consider the IF-THEN rules provided by the model to be rigorous and consistent with expert knowledge. In the relatively simple CCPP dataset, almost every rule follows the basic principles: AT and V are negatively correlated with energy output, while AP and RH are positively correlated. In the more complex MHR dataset, each rule can also be reasonably interpreted based on relevant medical literature. This provides strong evidence that KANFIS does more than merely fit statistical distributions; it successfully rediscovers the underlying physical or physiological laws governing the data in an unsupervised manner. For higher-dimensional datasets, KANFIS can also control the number of features included in each rule through regularization.

### 4.3. Ablation Experiments

The proposed regularization mechanism is designed to exert precise control over the model structure. By applying regularization, we aim to ensure that the features used in the rules remain within a reasonable range, thereby maximizing interpretability. To validate this, we conducted ablation studies on five datasets to investigate the impact of regularization on model behavior. As shown in Figure 3, the regularization term plays a crucial role in controlling model complexity and promoting interpretability. In the absence of regularization, rules across all datasets tend to incorporate all available features, leading to dense and entangled logic that is difficult to interpret. Introducing regularization enforces implicit feature selection at the rule level, substantially reducing the average number of active features per rule. Consequently, the model learns rules that better align with domain knowledge, with each rule focusing exclusively on the most informative variables, thereby significantly enhancing interpretability. Our regularization method substantially reduces the number of features utilized by each rule, achieving a maximum reduction of 63.94% on the Spambase dataset. Importantly, despite this increase in sparsity, predictive performance remains largely preserved, suggesting that the proposed regularization effectively eliminates redundant or noisy connections without compromising critical information. The impact of regularization on model accuracy is presented in Table 4.

### 4.4. Complexity Analysis

In this section, we analyze the computational complexity of the proposed model. On conventional datasets, the complex-

*Table 4.* **Impact of Different Levels of Regularization on Model Accuracy.**

| Dataset | Metric | Normal Regularization | High Regularization | Low Regularization |
|---------|--------|----------------------|---------------------|--------------------|
| CCPP | RMSE | 4.1240 | 4.2304 | 4.1903 |
| Parkinsons | RMSE | 0.0397 | 0.0418 | 0.0403 |
| BCW | Acc | 0.9912 | 0.9474 | 0.9474 |

ity of KANFIS is comparable to that of ANFIS, while its training efficiency significantly surpasses that of KAN. However, as the dimensionality and scale of the datasets increase, traditional ANFIS suffers from the curse of dimensionality, and the training difficulty of KAN also grows substantially, ultimately making both ANFIS and KAN impractical for high-dimensional and large-scale scenarios. In contrast, KANFIS demonstrates clear advantages in both scalability and training efficiency. The corresponding comparisons of training time versus accuracy, as well as parameter count versus FLOPs, are provided in the **appendix E**.

## 5. Conclusion

This work revisits the classical ANFIS model from the perspective of the Kolmogorov-Arnold representation theorem and proposes a compact neuro-fuzzy architecture, termed KANFIS. Without introducing additional structural complexity, KANFIS combines additive functional decomposition with Interval Type-2 fuzzy modeling, achieving competitive predictive performance while preserving intrinsic interpretability and robustness to uncertainty through explicit fuzzy reasoning, as well as improved scalability in high-dimensional settings. More broadly, KANFIS offers a unifying perspective for integrating classical neuro-fuzzy models with modern function representation principles. Future work may explore the incorporation of richer fuzzy reasoning mechanisms or logical operators within this framework, as well as extensions to more complex tasks and application scenarios, further advancing interpretable neuro-symbolic learning.

## Impact Statement

This paper proposes KANFIS, a scalable neuro-symbolic framework designed for efficiency and interpretability. By avoiding the exponential growth of fuzzy rules in high-dimensional settings, KANFIS significantly reduces the structural complexity of traditional ANFIS and effectively alleviates the curse of dimensionality. The proposed architecture further improves model scalability, enabling efficient learning on larger and more complex datasets. In addition, this work introduces a rule regularization strategy that controls rule sparsity and distinctiveness, allowing the model to generate more compact, effective, and interpretable rule structures while maintaining competitive predictive performance.

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

# A. Universal Approximation Theorem of KANFIS

This section formally establishes the universal approximation capability of the KANFIS model. The proof demonstrates that by leveraging the inherent approximation power of its Interval Type-2 (IT2) Gaussian basis functions, the KANFIS architecture can approximate any continuous multivariate function on a compact domain.

## A.1. KANFIS Model Formulation

The single-layer KANFIS model is an additive network structure where the mapping from the input $\mathbf{x} = (x_1, \ldots, x_n)$ to the output $f_{\text{KANFIS}}(\mathbf{x})$ is defined by:

$$f_{\text{KANFIS}}(\mathbf{x}) = \sum_{j=1}^{R} w_j \cdot h_j(\mathbf{x}) + b$$

The rule activation $h_j(\mathbf{x})$ aggregates the feature contributions via summation: $h_j(\mathbf{x}) = \sum_{i=1}^{n} \phi_{ij}(x_i)$. The core component, the edge function $\phi_{ij}(x_i)$, is a summation of $M$ center type-reduced IT2 Gaussian MFs:

$$\phi_{ij}(x_i) = \sum_{m=1}^{M} \frac{1}{2} \left[ \mu_{U,ijm}(x_i) + \mu_{L,ijm}(x_i) \right]$$

## A.2. Key Lemma: RBF Approximation Power

The proof hinges on the ability of the parameterized edge functions to model single-variable functions.

**Lemma A.1.** *Let $\psi(x)$ be any continuous function defined on a compact set $C \subset \mathbb{R}$. For any $\epsilon_1 > 0$, the KANFIS edge function $\phi_{ij}(x)$ can be parameterized by adjusting its centers and widths $\{\mu, \sigma_l, \sigma_u\}$ such that it approximates $\psi(x)$ within the error tolerance $\epsilon_1$.*

$$\|\phi_{ij}(x) - \psi(x)\|_\infty < \epsilon_1$$

*Proof.* The structure of $\phi_{ij}(x)$ constitutes a linear combination of Gaussian basis functions with adjustable parameters. Based on the universal approximation theorem for Radial Basis Function Network (RBFN), this specific form of basis function superposition is sufficient to approximate any continuous single-variable function $\psi(x)$ on the compact domain $C$ to arbitrary precision. The IT2 nature enhances the model's capacity to handle uncertainty but retains the necessary mathematical property of basis function approximation. $\square$

## A.3. Universal Approximation Theorem

Theorem A.1 allows us to establish the main result for the overall KANFIS architecture.

**Theorem A.2.** *Let $K \subset \mathbb{R}^n$ be a compact set and $f : K \to \mathbb{R}$ be any continuous function. For any $\epsilon > 0$, there exists a KANFIS model $f_{KANFIS}(\mathbf{x})$ such that:*

$$\sup_{\mathbf{x} \in K} |f(\mathbf{x}) - f_{KANFIS}(\mathbf{x})| < \epsilon$$

*Proof.* The proof follows the established methodology for showing universal approximation of additive structures.

**1. Approximation by Ideal Additive Structure ($f^*$):** By the generalized universal approximation theorems for additive networks (e.g., TSK models), for any $\epsilon_2 > 0$, we assume the existence of an ideal function $f^*(\mathbf{x})$ with the KANFIS structure:

$$f^*(\mathbf{x}) = \sum_{j=1}^{R} c_j \cdot \left[ \sum_{i=1}^{n} \psi_{ij}(x_i) \right] + c_0$$

such that:

$$\|f(\mathbf{x}) - f^*(\mathbf{x})\|_\infty < \epsilon_2$$

**2. Edge Function Substitution and Error Control:** We construct the KANFIS model $f_K$ by replacing $\psi_{ij}$ with the learned edge functions $\phi_{ij}$, setting $w_j = c_j$ and $b = c_0$. Let $C_{\max} = \max_j |c_j|$. Based on Theorem A.1, we choose the

approximation error $\epsilon_1$ for $\phi_{ij}$ to satisfy the required global bound:

$$\|\psi_{ij}(x_i) - \phi_{ij}(x_i)\|_\infty < \epsilon_1 = \frac{\epsilon - 2\epsilon_2}{2nRC_{\max}}$$

**3. Bounding the Structural Error ($\|f^* - f_K\|_\infty$):** The error introduced by substituting $\psi_{ij}$ with $\phi_{ij}$ is bounded using the triangle inequality:

$$
\begin{aligned}
\|f^* - f_K\|_\infty &\leq \sum_{j=1}^{R} |c_j| \sum_{i=1}^{n} \|\phi_{ij}(x_i) - \psi_{ij}(x_i)\|_\infty \\
&< RC_{\max} n \cdot \epsilon_1 \\
&= RC_{\max} n \cdot \left( \frac{\epsilon - 2\epsilon_2}{2nRC_{\max}} \right) \\
&= \frac{\epsilon}{2} - \epsilon_2
\end{aligned}
$$

**4. Final Error Analysis:** The total approximation error is the sum of the ideal approximation error ($\epsilon_2$) and the structural error ($\|f^* - f_K\|_\infty$):

$$
\begin{aligned}
\|f - f_K\|_\infty &\leq \|f - f^*\|_\infty + \|f^* - f_K\|_\infty \\
&< \epsilon_2 + \left( \frac{\epsilon}{2} - \epsilon_2 \right) \\
&= \frac{\epsilon}{2}
\end{aligned}
$$

By selecting $\epsilon_2$ sufficiently small (e.g., $\epsilon_2 = \epsilon/4$), the total error is less than $\epsilon$. Since $\epsilon$ is arbitrary, the proof is complete. $\square$

## B. Comparison of Approximation Properties: Additive vs. Product Fuzzy Systems

This appendix provides a formal analysis comparing the function approximation capabilities and complexity scaling of Additive Fuzzy Systems (AFS, exemplified by KANFIS) and Product Fuzzy Systems (PFS, exemplified by ANFIS). Let $N$ be the input dimension and $X \subset \mathbb{R}^N$ be a compact input space.

### B.1. Structural Definitions

**Definition B.1** (Product Fuzzy System (PFS) Structure). The output $f_{\mathrm{prod}}(\mathbf{x})$ of a PFS (e.g., TSK-ANFIS) uses the Product T-Norm for rule aggregation. The firing strength $\tau_j(\mathbf{x})$ is:

$$\tau_j(\mathbf{x}) = \prod_{i=1}^{N} \mu_{i,j}(x_i)$$

The system output is calculated by the weighted sum of consequents:

$$f_{\mathrm{prod}}(\mathbf{x}) = \frac{\sum_{j=1}^{R} \tau_j(\mathbf{x}) \cdot c_j}{\sum_{j=1}^{R} \tau_j(\mathbf{x})}$$

where $R$ is the number of rules, and $c_j$ are the consequent parameters.

**Definition B.2** (Additive Fuzzy System (AFS) Structure). The output $f_{\mathrm{sum}}(\mathbf{x})$ of an AFS (e.g., KANFIS) is defined by an additive decomposition, following the structure of a Kolmogorov-Arnold Network:

$$f_{\mathrm{sum}}(\mathbf{x}) = \sum_{j=1}^{H} \left( \sum_{i=1}^{N} \psi_{i,j}(x_i) \right)$$

where $H$ is the hidden layer width, and $\psi_{i,j}(x_i)$ is the single-variable function derived from the input $x_i$ to the hidden node $j$.

**B.2. Approximation Capability and Scalability Disparity**

**Theorem B.3** (Universal Approximation Equivalence)**.** *Both Product Fuzzy Systems $f_{prod}$ and Additive Fuzzy Systems $f_{sum}$ are universal function approximators on a compact set $X$.*

*Proof.* 1. **PFS ($f_{\text{prod}}$):** Established by the fact that the normalized basis functions of a TSK system form a partition of unity, allowing them to uniformly approximate any continuous function on a compact set (Stone-Weierstrass Theorem).

2. **AFS ($f_{\text{sum}}$):** The structure is a two-layer KAN, which is a universal approximator based on the properties derived from the Kolmogorov-Arnold Representation Theorem. The use of continuous and trainable $\psi_{i,j}(x_i)$ satisfies the necessary conditions for this property.

Thus, their theoretical ability to approximate any function is equivalent. $\qquad\square$

**Theorem B.4** (Efficiency Disparity in High Dimensions)**.** *When approximating functions $f : X \to \mathbb{R}$, the parameter complexity of the Additive Fuzzy System $f_{sum}$ scales linearly with the dimension $N$, while the complexity of the Product Fuzzy System $f_{prod}$ scales exponentially with $N$.*

*Proof.* Let $M$ be the number of MFs per dimension ($M \geq 2$), and $P_\psi$ be the parameter count for a single function $\psi_{i,j}$.

1. **Complexity of PFS ($f_{\text{prod}}$):** A complete rule base requires $R = M^N$ rules. The total parameter complexity $P_{\text{prod}}$ is:

$$P_{\text{prod}} \propto O(M^N)$$

This exponential scaling severely limits the practical application of PFS in high-dimensional space ($N$).

2. **Complexity of AFS ($f_{\text{sum}}$):** The total number of single-variable functions $\psi_{i,j}$ is $N \cdot H$. The total parameter complexity $P_{\text{sum}}$ is:

$$P_{\text{sum}} \propto O(N \cdot H \cdot P_\psi)$$

This demonstrates linear scaling with respect to the input dimension $N$.

Due to the exponential vs. linear scaling, $\lim_{N \to \infty} P_{\text{prod}}/P_{\text{sum}} = \infty$. Consequently, the Additive Fuzzy System (KANFIS) offers superior practical scalability and efficiency for complex, high-dimensional function approximation. $\square$ $\qquad\square$

# C. T1 Membership Function Specifications

This appendix provides the explicit definitions of the Type-1 membership functions used in this work. These formulations complement the Gaussian case presented in the main text and are included here for completeness and reproducibility.

**C.1. Structural Definitions**

**Definition C.1** (Generalized Bell Membership Function)**.** The Generalized Bell membership function defines a smooth and localized fuzzy response for an input variable $x_i$. It is parameterized as

$$\mathcal{M}_{i,j,k}(x_i) = \frac{1}{1 + \left| \frac{x_i - c_{i,j,k}}{a_{i,j,k}} \right|^{2b_{i,j,k}}}$$

,

where $a_{i,j,k} > 0$ controls the width of the membership region, $b_{i,j,k} > 0$ adjusts the slope around the center, and $c_{i,j,k}$ denotes the center location. All parameters are learnable.

**Definition C.2** (Sigmoid Membership Function)**.** The Sigmoid membership function models monotonic fuzzy relationships and is defined as

$$\mathcal{M}_{i,j,k}(x_i) = \frac{1}{1 + \exp\left(-\alpha_{i,j,k}(x_i - \beta_{i,j,k})\right)}$$

,

where $\alpha_{i,j,k}$ controls the slope and direction of monotonicity, and $\beta_{i,j,k}$ specifies the transition center. Both parameters are learnable.

## C.2. Functional Characteristics

The Generalized Bell function provides symmetric, compactly supported responses with adjustable tail behavior, enabling flexible modeling of localized nonlinearities. In contrast, the Sigmoid function introduces asymmetric and monotonic responses, which are suitable for representing threshold-like or directional dependencies. Together with the Gaussian membership function described in the main text, these Type-1 membership functions form a complementary set of basis functions for capturing heterogeneous nonlinear structures.

## D. Supplementary Datasets and Baseline Models for Comparison

As shown in Table 5 and Table 6, we further enhance the credibility of this work by introducing three additional datasets and three supplementary baseline models for comparison. The added datasets are all either high-dimensional or large-scale datasets, which can effectively validate whether KANFIS truly overcomes the dimensional limitations commonly encountered in traditional ANFIS models. The additional baseline models are representative interpretable models. Compared with these interpretable approaches, KANFIS achieves superior predictive accuracy. Moreover, during the experiments, we observed that these baseline models exhibit substantially higher computational complexity and training difficulty than KANFIS.

*Table 5.* **Accuracy of KANFIS on the supplementary datasets.** The three additional datasets reported in the table are all highly high-dimensional datasets. Among them, the ForestCover dataset was specifically included due to its extremely large scale, containing 581,012 instances.

| Dataset | Dimension | T1-KANFIS (Acc) | IT2-KANFIS |
|---|---|---|---|
| APS Failure at Scania Trucks | 171 | 0.9898 | 0.9900 |
| Dexter | 20000 | 0.8667 | 0.9333 |
| ForestCover | 54 | 0.9420 | 0.9491 |

*Table 6.* **Performance comparison with other interpretable models.** The table presents comparative results with three representative interpretable models, namely NAMs, GAMI-NET, and NODE-GAM. Due to limitations in the available open-source implementation, NODE-GAM was unable to produce results on the Parkinsons dataset.

| Dataset | Metric | Ours | NAMs | GAMI-NET | NODE-GAM |
|---|---|---|---|---|---|
| Parkinsons | RMSE | 0.0397 | 0.0408 | 0.0403 | – |
| BCW | Acc | 0.9912 | 0.9651 | 0.9649 | 0.9737 |

## E. Comparison of Training Time, Accuracy, Parameters, and FLOPs

To further evaluate the computational efficiency and scalability of KANFIS, we provide additional comparisons of training time, accuracy, parameter count, and FLOPs in Table 7 and Table 8 . These results further demonstrate that KANFIS maintains favorable computational efficiency while achieving strong predictive performance, particularly in high-dimensional and large-scale data scenarios.

*Table 7.* **Time-Accuracy Comparison of Models.** The training time is measured in seconds per training epoch. For regression datasets, accuracy is measured using RMSE, while for classification datasets, accuracy is measured using Acc. Values are reported as RMSE and Accuracy.

| Dataset | IT2-KANFIS | T1-KANFIS | MLP | ANFIS | KAN |
|---|---|---|---|---|---|
| CCPP | 0.3914, 4.1240 | 0.3560, 3.9542 | 0.0902, 4.1883 | 0.3042, 3.9980 | 4.8063, 3.9313 |
| BCW | 0.0212, 0.9912 | 0.0207, 0.9912 | 0.0067, 0.9912 | – | 1.4943, 0.9736 |
| Spam | 0.1357, 0.9392 | 0.1175, 0.9381 | 0.0475, 0.9175 | – | – |

*Table 8.* **Comparison of Parameter Count and FLOPs for Different Models.** Values are reported as the number of parameters and FLOPs.

| Dataset | IT2-KANFIS | T1-KANFIS | MLP | ANFIS | KAN |
|---|---|---|---|---|---|
| CCPP | 521, 2600 | 361, 1640 | 193, 480 | 429, 1274 | 1600, 9972 |
| BCW | 4822, 24330 | 3986, 18396 | 1058, 2208 | 5139, 15398 | 2240, 11518 |
| Spam | 7314, 36960 | 5034, 23280 | 1922, 3936 | – | – |

