# OpenReview forum: "KANFIS: A Neuro-Symbolic Framework for Interpretable and Uncertainty-Aware Learning"
_ICML.cc/2026/Conference — ICML 2026 regular_

### Official Review · Reviewer_UTbc · 2026-03-08

**Soundness:** 3
**Presentation:** 3
**Significance:** 2
**Originality:** 3
**Overall Recommendation:** 3
**Confidence:** 4

**Summary:**

This paper proposes KANFIS, a neuro-fuzzy architecture that combines ANFIS-style fuzzy reasoning with Kolmogorov-Arnold-style additive function decomposition. The key idea is to replace conventional product-based fuzzy aggregation, which leads to exponential rule growth, with additive aggregation over learnable fuzzy functional edges, so that complexity scales linearly with input dimensionality. The model supports both Type-1 and Interval Type-2 fuzzy logic, and adds sparsity and rule-distinctiveness regularization to promote compact, interpretable rule sets. Empirically, the paper compares KANFIS against MLP, ANFIS variants, and KAN on five datasets, and includes two qualitative interpretability case studies plus a regularization ablation.

**Compliance With Llm Reviewing Policy:**

Affirmed.

**Final Justification:**

Strengths:
I find the topic of this paper interesting, as it addresses a meaningful and well-motivated problem. Classical ANFIS models are inherently interpretable, but they notoriously suffer from rule explosion in higher-dimensional settings. The paper proposes a clean architectural response by reinterpreting KAN-style learnable univariate functions as fuzzy membership functions and aggregating them additively rather than multiplicatively. This idea is intuitive and technically coherent.

Weaknesses:
1. The main weakness of the paper is the limited empirical evaluation. The experiments are conducted on only five datasets, and the set of competitive baselines is relatively narrow. Moreover, the proposed method does not achieve the best performance across all benchmarks.
2. Another concern is the paper’s overall fit for ICML. In my view, the topic and contribution seem more aligned with a fuzzy-systems venue than with ICML. This is not to say that the paper is not good, but rather that its scope and emphasis may be less compelling for the ICML audience.
3. The paper’s uncertainty-aware claim is under-validated. The method supports IT2 fuzzy logic and motivates this as enabling uncertainty modeling, but there is no dedicated evaluation of uncertainty quality, such as calibration, robustness to noise, ambiguity handling, OOD behavior, or any downstream decision-making benefit from uncertainty estimates. In fact, the reported results do not clearly show that IT2-KANFIS is consistently better than T1-KANFIS; on CCPP and Parkinsons, the T1 variant is slightly better on all three reported regression metrics.

**Key Questions For Authors:**

1. Why is the empirical evaluation limited to only five datasets and a relatively narrow set of baselines?
2. What is the main practical advantage of KANFIS when it does not consistently achieve the best predictive performance?
3. Can the authors provide a dedicated evaluation of the claimed uncertainty-awareness, such as calibration or OOD uncertainty quality?
4. How should readers interpret the IT2 extension given that IT2-KANFIS does not consistently outperform T1-KANFIS?

**Limitations:**

yes

**Strengths And Weaknesses:**

Strengths:
I find the topic of this paper interesting, as it addresses a meaningful and well-motivated problem. Classical ANFIS models are inherently interpretable, but they notoriously suffer from rule explosion in higher-dimensional settings. The paper proposes a clean architectural response by reinterpreting KAN-style learnable univariate functions as fuzzy membership functions and aggregating them additively rather than multiplicatively. This idea is intuitive and technically coherent.

Weaknesses:
1. The main weakness of the paper is the limited empirical evaluation. The experiments are conducted on only five datasets, and the set of competitive baselines is relatively narrow. Moreover, the proposed method does not achieve the best performance across all benchmarks.
2. Another concern is the paper’s overall fit for ICML. In my view, the topic and contribution seem more aligned with a fuzzy-systems venue than with ICML. This is not to say that the paper is not good, but rather that its scope and emphasis may be less compelling for the ICML audience.
3. The paper’s uncertainty-aware claim is under-validated. The method supports IT2 fuzzy logic and motivates this as enabling uncertainty modeling, but there is no dedicated evaluation of uncertainty quality, such as calibration, robustness to noise, ambiguity handling, OOD behavior, or any downstream decision-making benefit from uncertainty estimates. In fact, the reported results do not clearly show that IT2-KANFIS is consistently better than T1-KANFIS; on CCPP and Parkinsons, the T1 variant is slightly better on all three reported regression metrics.

---

> ### Author Rebuttal · Authors · 2026-03-30
>
> **W1 & Q1:** Datasets and Baseline Models
>
> **A1:** We argue that the five datasets we selected already provide sufficient evidence that the KANFIS model outperforms traditional ANFIS and KAN models on high-dimensional datasets. Specifically, on the Spambase and BCW datasets, both ANFIS and KAN face training difficulties. Additionally, the baseline models we selected for comparison are the ones most similar in type to our proposed model. Although NAMs and GAMI-Net are also explainable models, their interpretability mechanisms differ from KANFIS, which makes them less comparable. In response to your feedback, we have added additional datasets and baseline models for comparison, with the results presented in Tables 1 and 2. The new experimental data clearly demonstrate that our proposed architecture enables ANFIS to fully adapt to high-dimensional and large-scale environments, and KANFIS achieves superior accuracy compared to other models on most datasets.
>
> |Dataset|Dimension|T1-KANFIS ( Acc ) |IT2-KANFIS|
> |-|-|-|-|
> |APS Failure at Scania Trucks|171|0.9898|0.99|
> |Dexter|20,000|0.8667|0.9333|
> |ForestCover|54|0.942|0.9491|
>
> Table 1: Accuracy of KANFIS on Three Additional High-Dimensional and Large-Scale Datasets
>
> | Dataset   | |Ours| NAMs| GAMI-NET | NODE-GAM |
> |-|-|-|-|-|-|
> | Parkinsons | RMSE | **0.0397** |0.0408 | 0.0403 | |
> | BCW | Acc | **0.9912**|0.9651 | 0.9649 | 0.9737 |
>
> Table 2: Comparison with Additional White-Box Models. Due to incomplete open-source availability and time constraints, some models do not perform well on certain datasets.
>
> **W2:** Classification of the Paper’s Domain
>
> **A2:** We respectfully disagree with the concern regarding the suitability for ICML. While KANFIS is inspired by fuzzy neural systems, its core contribution is a scalable and interpretable learning framework that addresses the dimensionality explosion issue in high-dimensional settings, which is central to modern machine learning. Prior ICML works have also explored related paradigms [1, 2].
>
> **W3 & Q3:** Uncertainty Analysis
>
> **A3:** IT2-KANFIS inherits the characteristics of IT2-ANFIS, naturally providing the ability for uncertainty modeling. The capability of IT2 for uncertainty modeling is described in detail in papers [1, 2], but due to space limitations, we have not elaborated on this aspect in the paper. we will further include a more detailed discussion in the appendix. In Table 3, we present the standard deviation of accuracy obtained from training the model three times with random seeds, using the Spambase dataset as an example.
>
> | | IT2-KANFIS  | T1-KANFIS  | MLP| NAMs|GAMI-Net
> |-|-|-|-|-|-|
> | Standard Deviation of Acc | 0.002345    | 0.007594   | 0.00602 | 0.00942 |0.01009
>
> Table 3: Standard Deviation of Accuracy for Each Model under Random Seed Conditions.
>
> **Q2:** Main Advantages of KANFIS
>
> **A4:** The main contribution of this paper lies in addressing the dimensionality explosion problem faced by traditional ANFIS models on high-dimensional datasets, reducing the complexity of ANFIS from exponential growth with respect to dimensionality to linear growth. Furthermore, KANFIS not only utilizes efficient computational methods but also optimizes the interpretability of fuzzy neural network rules. Importantly, this design, which enhances training efficiency and provides high-quality, interpretable rules, does not compromise the model's performance.
>
> **Q4:** IT2 Extension
>
> **A5:** IT2-KANFIS and T1-KANFIS each have distinct advantages in different application scenarios. On datasets with high disturbance, IT2-KANFIS typically yields more convincing results than T1-KANFIS, while on simpler datasets, IT1-KANFIS tends to achieve higher accuracy. This is entirely reasonable. As seen with traditional ANFIS, IT2-ANFIS is not always superior to T1-ANFIS across all datasets, as evidenced by both the literature [ 3, 4 ] and our experimental data. Overall, we believe that the IT2 extension broadens the applicability of the KANFIS model, making it more meaningful for research in a wider range of scenarios.
>
> [ 1 ] Deep Fuzzy Multi-view Learning for Reliable Classification
>
> [ 2 ] Human Cognition-Inspired Hierarchical Fuzzy Learning Machine
>
> [ 3 ] Type-2 fuzzy sets and systems: an overview
>
> [ 4 ] Interval type-2 fuzzy logic systems made simple

---

> > ### Author Rebuttal · Reviewer_UTbc · 2026-04-03
> >
> > Thank you for the detailed rebuttal and additional experimental results. The clarifications on the expanded datasets, additional baselines, and the practical motivation of KANFIS are helpful. I appreciate the authors’ effort in addressing my concerns regarding empirical scope and the role of the IT2 extension. I will consider adjusting my score after discussing with the other reviewers.

---

> > > ### Author Response · Authors · 2026-04-06
> > >
> > > We sincerely thank you for the positive evaluation of our work and for considering an adjustment of the score after the discussion. We will carefully address the identified limitations and further improve the manuscript in the final version. Once again, we greatly appreciate your constructive feedback and valuable guidance.

---

### Official Review · Reviewer_1TSy · 2026-03-11

**Soundness:** 3
**Presentation:** 3
**Significance:** 4
**Originality:** 4
**Overall Recommendation:** 5
**Confidence:** 3

**Summary:**

This paper aims to broadly address the drawbacks of adaptive neuro-fuzzy inference systems (ANFIS), which are powerful hybrid models that combine the learning capabilities of neural network and the human like reasoning of fuzzy logic. One major issue with ANFIS is that they are structurally complex. Therefore, this paper explored the concept of neuro-fuzzy Kolmogorov-Arnold Networks. They develop Kolmogorov-Arnold Neuro-Fuzzy Inference Systems (KANFIS), which aims to unify the fuzzy reasoning with adaptive function decomposition. They demonstrate through literature, the benefit and potential advantages of replacing the fixed linear transformations with learnable nonlinear operators, especially for fuzzy inference models.

**Compliance With Llm Reviewing Policy:**

Affirmed.

**Final Justification:**

I maintain my Accept recommendation. The paper proposes KANFIS, a novel integration of neuro-fuzzy inference with Kolmogorov-Arnold-style additive architectures. This is an interesting and potentially impactful direction for reducing ANFIS complexity while preserving interpretability through explicit rule-based reasoning. The methodology is technically sound and supported by empirical comparisons.

My main concerns were related to computational overhead and the lack of formal interpretability evaluation. The authors clarified that KANFIS adopts the additivity principle without spline-based activations, which significantly reduces training complexity. They also provided parameter count, FLOPs, and training time comparisons to support this claim. Regarding interpretability, the authors justified rule validation using domain knowledge, which is common practice in fuzzy logic research, and agreed to include limitations in the final version.

Overall, the rebuttal adequately addressed my concerns but did not substantially change my evaluation. I therefore maintain my Accept recommendation.

**Key Questions For Authors:**

1.KANs are fundamentally slower to train than the traditional multi-layer perceptrons (MLPs), often taking 10 times longer. Furthermore, even though a combined model like KANFIS solves the exponential growth rule of ANFIS, the underlying reliance on KAN’s complex, learnable spline-based activation function will still drastically increase the overall parameter count, compared to simpler models. How did you handle such extreme computational bottleneck?
2. One major selling point of combining fuzzy logic with KANs is enhanced interpretability/transparency. However, many studies lack formal mathematical metrics to quantify and prove this interpretability, mostly depending on qualitative visual inspections of the extracted rules and splines. How has/can this paper address this lack of formal interpretability metrics?
It would be helpful when comparing the hybrid model against baselines, to look at the ratio of parameters to training time. This is because KANs lack efficient batching, it is important to verify if the reduction in fuzzy rules actually translates to faster convergence or if the spline calculations negate those efficiency gains.

**Limitations:**

While combining KANs with fuzzy logic has exciting theoretical advantages for accuracy and interpretability, there are practical limitations and challenges that could arise. It will be insightful to highlight the limitations of this hybrid model in the paper.

**Strengths And Weaknesses:**

The concept of combining the reasoning of ANFIS within a Kolmogorov-Arnold Style additive architecture is quite interesting and novel. There was also a detailed theoretical detail on the concepts of KANFIS.

---

> ### Author Rebuttal · Authors · 2026-03-30
>
> **Q1:** Model Complexity
>
> **A1:** KANFIS does not directly adopt the learnable spline activation functions used in KAN. Instead, it leverages the additivity principle of KAN to reconfigure ANFIS. As a result, KANFIS does not face the extreme training difficulties that KAN encounters. We have summarized the training time, model parameters, and FLOPs in Table 1. The results indicate that the complexity of KANFIS is much lower than that of KAN, and KANFIS demonstrates relatively high computational efficiency. Additionally, during our training process, we found that typically only a single layer of KANFIS is needed to achieve the best performance. Therefore, no extreme computational bottlenecks were encountered during actual training.
>
> | Dataset | IT2-KANFIS | T1-KANFIS | ANFIS | KAN |
> |-|-|-|-|-|
> | CCPP  | 521, 2600, 0.3914 | 361, 1640, 0.356 | 429, 1274, 0.3042 | 1600, 9972, 4.8063 |
> | BCW | 4822, 24330, 0.0212 | 3986, 18396, 0.0207 | - | 9920, 30070, 1.4943 |
> | Spam | 7314, 36960, 0.1357 | 5034, 23280, 0.1175 | - | - |
>
> Table 1: Model Parameter Count, FLOPs, and Training Time.
>
> **Q2:** Interpretability Evaluation
>
> **A2:** The model proposed in this paper provides clear and explicit IF-THEN rules, which are typically difficult to assess using traditional quantitative metrics. To address this issue, we validate and verify the correctness of the model-generated rules by comparing them with conclusions from relevant domain-specific literature. We understand that expert evaluation is one of the most commonly used methods for validating interpretability in explainable AI research, as seen in papers [ 1, 2, 3 ], where this method is used to verify the rules. Furthermore, we believe that the IF-THEN rules generated by KANFIS align closely with human reasoning patterns, making this method particularly well-suited for validating interpretability.
>
> [ 1 ] Critical thinking about explainable AI ( XAI ) for rule-based fuzzy systems
>
> [ 2 ] Stop explaining black box machine learning models for high stakes decisions and use interpretable models instead
>
> [ 3 ] Interpreting black-box models: a review on explainable artificial intelligence

---

> > ### Author Rebuttal · Reviewer_1TSy · 2026-04-03
> >
> > 1. The clarification that KANFIS utilizes the additivity principle of KAN without the spline-based activation functions is a crucial distinction. This explains the significantly lower FLOPs and training times compared to standard KANs and successfully addresses my concerns regarding computational bottlenecks.
> > 2. Regarding interpretability, while I acknowledge the authors' point that IF-THEN rules are inherently more transparent to human experts, I still believe the field would benefit from more formal quantification. However, I accept that qualitative validation against domain literature is a standard practice in fuzzy logic research.
> > 3. I recommend the authors include the data from Table 1 in the final manuscript and add a brief discussion on the 'Limitations' of the model (specifically regarding the trade-offs of opting out of spline-based activations) as discussed.
> >
> > I maintain my recommendation.

---

> > > ### Author Response · Authors · 2026-04-06
> > >
> > > We sincerely thank you for the recognition and valuable feedback on our work. We will include additional analyses of model complexity, specifically parameter count, FLOPs, and training time in the appendix. Furthermore, we will add a discussion on the limitations of the proposed model in the manuscript, particularly regarding the trade-offs associated with not using spline-based activation functions.

---

### Official Review · Reviewer_ddT9 · 2026-03-11

**Soundness:** 2
**Presentation:** 2
**Significance:** 2
**Originality:** 3
**Overall Recommendation:** 4
**Confidence:** 3

**Summary:**

The paper proposes KANFIS, an architecture that extends Adaptive Neuro-Fuzzy Inference System (ANFIS) using the Kolmogorov-Arnold representation theorem. The key idea is to replace the product-based rule aggregation in ANFIS with an additive aggregation mechanism where learnable membership functions serve as the univariate functions in a KAN-style decomposition. To this end, the model supports both Type-1 and Interval Type-2 fuzzy logic, maintaining interpretability. Experiments are conducted on five datasets comparing against MLP, ANFIS, and KAN baselines.

**Compliance With Llm Reviewing Policy:**

Affirmed.

**Final Justification:**

The paper demonstrates the idea of combining KAN and ANFIS to overcome their limitations. The experiments support this idea. The set of datasets and baselines is not very diverse, so it may lack an impact on the general ML community.

**Key Questions For Authors:**

- How does KANFIS perform against the interpretable baselines discussed in Section 2.1 (e.g., NAMs, NODE-GAM, GAMI-Net) on standard tabular benchmarks?
- Is there any qualitative metrics to validate the scalability (e.g., comparisons on wall-clock training time, memory usage, etc) of the proposed method?
- For the multi-layer configuration, how are the interpretable rules extracted, and does interpretability degrade with depth?
- What is the sensitivity of performance and sparsity to the hyperparameters $λ_s$ and $λ_d$?

**Limitations:**

No. The paper lacks an explicit limitations discussion. It is encouraged to include discussions, e.g., on the small-scale experiments (max 57 features), the absence of recently established interpretable baselines, the lack of quantitative interpretability evaluation, and no hyperparameter sensitivity analysis.

**Strengths And Weaknesses:**

## Strengths
- The proposed architecture is well-motivated. The connection between KAN's learnable univariate functions and fuzzy membership functions is elegant and theoretically grounded. The additive aggregation resolving the curse of dimensionality in ANFIS is a meaningful contribution.
- The introduced regularization methods for interpretability are interesting contribution. Figure 3 convincingly shows the regularization reduces features per rule significantly.
- The paper presents meaningful interpretability case studies. Tables 2 and 3 provide concrete extracted rules with domain-grounded interpretations for CCPP and MHR datasets. The rules align with known physical and medical principles, which is a nice qualitative validation that the model captures meaningful structure rather than spurious correlations.
- The proposed method is theoretically backed up via proof in the appendix.

## Weaknesses
- The experimental evaluation is limited and unconvincing, which is the most significant weakness. Only five datasets are used, most of which are small and low-dimensional (CCPP has 4 features, Parkinsons has ~20, BCW has 30, MHR has 7). For a paper whose central claim is scalability to high dimensions, the evaluation needs datasets with hundreds (or thousands) of features. The lack of standard tabular benchmarks with sufficiently large feature sets weakens the empirical contribution considerably.
- Related to the above, the paper compares only against MLP, ANFIS variants, and KAN.  However, it is not clear why beating those baselines is sufficient to support the paper's claim, as there is no justification.
- The interpretability evaluation is only qualitative, and no quantitative metrics are available. The authors manually inspect rules and verify them against domain knowledge. The claim of *intrinsic interpretability* may require more rigorous validation.
- The scalability claims are not empirically supported at scale. The theoretical linear scaling argument is sound, but there are no wall-clock time comparisons, no memory profiling, and no experiments on genuinely high-dimensional data. The paper argues KAN failed on Spambase due to computational cost, but doesn't quantify this or show how KANFIS scales in comparison.
- There is a misattributed domain citation that undermines interpretability validation. In Table 3, the rule "DiastolicBP is LOW → high cardiovascular risk" is cited to Amin et al. (2006), a paper exclusively about estradiol, testosterone, and hip fracture risk in elderly men, and unrelated to blood pressure or cardiovascular disease. Any clarification will be beneficial.

---

> ### Author Rebuttal · Authors · 2026-03-30
>
> **W1:** Datasets
>
> **A1:** The five datasets we selected already include high-dimensional datasets. On the BCW and Spambase datasets, both ANFIS and KAN face significant training difficulties, whereas KANFIS is able to train smoothly. This is sufficient to demonstrate that KANFIS outperforms traditional KAN and ANFIS on high-dimensional datasets. In response to your question, we have conducted additional experiments on four high-dimensional, large-scale datasets, the results of which are presented in Table 1.
>
> |Dataset|Dimension|T1-KANFIS ( Acc )|IT2-KANFIS|
> |-|-|-|-|
> | APS Failure at Scania Trucks|171|0.9898|0.99|
> | Dexter | 20,000 | 0.8667|0.9333|
> | ForestCover |54|0.942|0.9491|
>
> Table 1: Accuracy of KANFIS on Three Additional High-Dimensional and Large-Scale Datasets
>
> **W2 & Q1:** Comparison with Baselines
>
> **A2:** We believe that KAN and ANFIS are the most comparable baseline models to the proposed method, while models such as NAMs and GAMI-Net differ significantly in terms of interpretability from KANFIS. Therefore, they were not included in the comparison. In response to your comment, we have added a comparison with three explainable models—NAMs, GAMI-Net, and NODE-GAM—which is presented in Table 2.
>
> | Dataset   | |Ours| NAMs| GAMI-NET | NODE-GAM |
> |-|-|-|-|-|-|
> | Parkinsons | RMSE | 0.0397 |0.0408 | 0.0403 | |
> | BCW | Acc | 0.9912|0.9651 | 0.9649 | 0.9737 |
>
> Table 2: Comparison with Additional White-Box Models. Due to incomplete open-source availability and time constraints, some models do not perform well on certain datasets.
>
> **W3:** Interpretability Evaluation
>
> **A3:** In the field of explainable AI ( XAI ), expert evaluation is often considered the most important basis for validating model interpretability. The relevant explainable AI papers [1, 2, 3 ] also adopt this approach for interpretability assessment. To ensure a fair comparison, this paper follows the same method. Moreover, the IF-THEN rules generated by KANFIS align with human cognitive patterns, making it particularly suitable for expert evaluation as a validation method.
>
> **W4 & Q2:** Scalability
>
> **A4:** We have summarized the training time, parameter count, and FLOPs of KANFIS in Table 3 for reference. The results show that, compared to other explainable models of similar type, the model complexity of KANFIS is acceptable. Moreover, KANFIS maintains high computational efficiency without compromising its interpretability or accuracy. Therefore, we believe that KANFIS has the potential for application in large-scale scenarios.
>
> | Dataset | IT2-KANFIS | T1-KANFIS | ANFIS | KAN |
> |-|-|-|-|-|
> |CCPP|521, 2600, 0.3914|361, 1640, 0.356| 429, 1274, 0.3042|1600, 9972, 4.8063|
> |BCW|4822, 24330, 0.0212|3986, 18396, 0.0207|-|9920, 30070, 1.4943|
> |Spam|7314, 36960, 0.1357|5034, 23280, 0.1175|-|-|
>
> Table 3: Model Parameter Count, FLOPs, and Training Time.
>
> **W5:** Citation Error
>
> **A5:** Thank you for pointing out the citation discrepancy in Table 3. During the compilation of references, we mistakenly cited McEvoy et al. (2016) [ 4 ] as Amin et al. (2006). The former is the key paper that discusses how low diastolic blood pressure leads to insufficient perfusion, which in turn causes myocardial damage.
>
> **Q3:** Multi-Layer Configuration
>
> **A6:** When the model consists of a single layer, explicit rules can be easily extracted. As the number of layers increases, higher layers are viewed as "rules controlling the rules," where the rules in higher layers typically determine whether the lower-layer rules are triggered. Although interpretability decreases with the increase in layers, our experiments show that the majority of datasets perform best with a KANFIS model containing only one layer. Even for the most complex datasets, the optimal performance can be achieved with a two-layer model.
>
> **Q4:** Regularization
>
> **A7:** The model's performance is not significantly affected by the hyperparameters 𝜆𝑠 and 𝜆𝑑, as shown in Table 4. The purpose of designing 𝜆𝑠 and 𝜆𝑑 is to control the rules output by the model, maintaining sparsity in the features of the rules and distinctiveness between the rules.
>
> | Dataset| KANFIS with Normal Regularization | KANFIS with High Regularization | KANFIS with Low Regularization |
> |-|-|-|-|
> | CCPP | RMSE  | 4.124 | 4.2304 | 4.1903 |
> | Parkinsons | RMSE  | 0.0397 | 0.0418 | 0.0403 |
> | BCW | Acc | 0.9912 | 0.9474 | 0.9474 |
>
> Table 4: Impact of Different Levels of Regularization on Model Accuracy
>
> [ 1 ] Critical thinking about explainable AI ( XAI ) for rule-based fuzzy systems
>
> [ 2 ] Stop explaining black box machine learning models for high stakes decisions and use interpretable models instead
>
> [ 3 ] Interpreting black-box models: a review on explainable artificial intelligence
>
> [ 4 ] Diastolic blood pressure, subclinical myocardial damage, and cardiac events: implications for blood pressure control

---

> > ### Author Rebuttal · Reviewer_ddT9 · 2026-04-03
> >
> > I thank the authors for the thorough rebuttal. It clarified the paper's core contribution: overcoming the limitations of KAN and ANFIS by combining them. The empirical results in the paper support this claim (although the set of baelines and datasets may not be very diverse, which may limit their impact on the community). As my major concerns are resolved, I will adjust my score.

---

> > > ### Author Response · Authors · 2026-04-06
> > >
> > > Thank you for recognizing the contributions and insights of our work, and for providing valuable feedback. We will incorporate the suggested additional datasets and baseline comparisons into the main paper or appendix, and further refine and strengthen our work in the final version.

---

### Official Review · Reviewer_Q6nn · 2026-03-12

**Soundness:** 3
**Presentation:** 4
**Significance:** 3
**Originality:** 3
**Overall Recommendation:** 4
**Confidence:** 4

**Summary:**

This paper introduces KANFIS (Kolmogorov-Arnold Neuro-Fuzzy Inference System), a novel neuro-symbolic architecture that addresses the scalability and complexity issues of traditional Adaptive Neuro-Fuzzy Inference Systems (ANFIS). By integrating the additive function decomposition principle from Kolmogorov-Arnold Networks (KAN) with fuzzy reasoning, the authors propose a model where rule complexity scales linearly rather than exponentially with input dimensionality. The framework supports both Type-1 and Interval Type-2 fuzzy logic to model uncertainty and utilizes entropy-based sparsity and diversity regularization to maintain compact, interpretable "IF-THEN" rules.

**Compliance With Llm Reviewing Policy:**

Affirmed.

**Key Questions For Authors:**

Q1：How does the training time and memory consumption of KANFIS compare to a standard MLP or a traditional ANFIS of equivalent predictive power? Since each edge in KANFIS involves learnable univariate functions, there may be a significant computational overhead. Providing a "Time-vs-Accuracy" or "Parameters-vs-FLOPs" analysis would clarify if the structural compactness translates into actual training efficiency, which is critical for large-scale applications.

Q2：How does the model perform when the input dimension $d$ exceeds 100 or 1000? In such cases, does the additive nature of the Kolmogorov-Arnold decomposition still capture sufficient feature interactions? Evidence of performance in higher-dimensional regimes would elevate the paper's significance from a specialized improvement to a broad architectural breakthrough.

Q3：The effectiveness of KANFIS in maintaining interpretability relies heavily on the entropy-based sparsity and rule distinctiveness terms. How sensitive is the model's performance and rule clarity to the tuning of these coefficients? If the model is highly sensitive, its "out-of-the-box" utility for practitioners may be limited. A brief discussion on the stability of these hyperparameters across the different datasets used would significantly strengthen the claim of robust interpretability.

**Limitations:**

yes

**Strengths And Weaknesses:**

Strengths

Soundness: The paper is technically grounded in the Kolmogorov-Arnold representation theorem, providing a rigorous mathematical basis for replacing fixed weights with learnable univariate fuzzy functions. The integration of Type-1 and Interval Type-2 fuzzy logic is methodologically sound, allowing the model to explicitly handle aleatoric uncertainty through Footprint of Uncertainty modeling. Furthermore, the authors include theoretical proofs for the model’s universal approximation capability and the validity of the additive aggregation scheme.

Presentation: The submission is well-structured and clearly written. It provides a logical flow from the limitations of current models to the proposed solution. Figure 2 effectively illustrates the architecture and computational flow, while the qualitative case study on the CCPP dataset demonstrates the model's transparency by mapping learned rules to established physical principles.

Significance: The work addresses a critical bottleneck in neuro-fuzzy systems: the "exponential explosion" of rules in high-dimensional spaces. By enabling rule complexity to scale linearly with input dimensionality, KANFIS makes neuro-symbolic learning more practical for real-world, high-stake applications like energy forecasting and finance.

Originality: The paper offers a novel synthesis of Kolmogorov-Arnold Networks and Neuro-Fuzzy Inference Systems. While KAN provides expressive power, it often lacks semantic clarity; KANFIS solves this by reinterpreting KAN’s edges as fuzzy membership functions, creating an intrinsically interpretable framework that extracts human-readable "IF-THEN" rules.

Weaknesses

Soundness: While the results on the five selected datasets are promising, the evaluation could be strengthened by including more complex, large-scale industrial datasets to further validate the claim of overcoming the curse of dimensionality in extreme high-dimensional regimes.

Presentation: The paper introduces specialized regularization terms, but it lacks a detailed sensitivity analysis or discussion on how difficult these are to tune across different domains to maintain the reported balance between accuracy and interpretability.

Significance: Although KANFIS is parameter-efficient, the use of learnable univariate functions (e.g., splines or Gaussian MFs) on every edge can increase computational complexity during the training phase compared to standard MLPs. The authors could provide more data on training time and resource consumption.

Originality: While the paper compares KANFIS to KAN and ANFIS, it would benefit from a more direct comparison or discussion regarding other recent Neural Additive Models or GAMI-Net to clearly delineate its unique advantages in capturing interactions versus existing additive architectures.

---

> ### Author Rebuttal · Authors · 2026-03-30
>
> **W1 & Q2:** Datasets
>
> **A1:** We appreciate your recognition of the performance of our model on the five selected datasets. We believe that these datasets are sufficient to demonstrate that KANFIS has successfully overcome the limitations of traditional ANFIS in high-dimensional spaces. On the BCW and Spambase datasets, ANFIS experiences dimensionality explosion, and the KAN model becomes increasingly difficult to train. In contrast, KANFIS maintains efficient training and high accuracy. To address the concerns you raised, we have expanded the evaluation to include four additional high-dimensional and large-scale datasets. The accuracy of KANFIS on these datasets is presented in Table 1.
>
> | Dataset | Dimension | T1-KANFIS ( Acc )  | IT2-KANFIS |
> |-|-|-|-|
> | APS Failure at Scania Trucks   | 171 | 0.9898 | 0.99 |
> | Dexter | 20,000 | 0.8667   | 0.9333 |
> | ForestCover | 54 | 0.942    | 0.9491 |
>
> Table 1: Accuracy of KANFIS on Three Additional High-Dimensional and Large-Scale Datasets
>
> **W2 & Q3:** Regularization Analysis
>
> **A2:** For the KANFIS model, the two regularization terms we designed primarily serve to adjust rule sparsity and distinctiveness, enhancing interpretability. As shown in Table 2, these parameters do not directly or significantly alter the model's performance. As a white-box model, users can guide hyperparameter tuning by analyzing the rules output by the model. This approach allows for rapid and stable adjustment to optimal performance, while maintaining the clarity and interpretability of the rules.
>
> | Dataset | Metric | Normal Regularization | High Regularization | Low Regularization |
> |-|-|-|-|-|
> | CCPP | RMSE   | 4.124 | 4.2304 | 4.1903 |
> | Parkinsons  | RMSE | 0.0397 | 0.0418 | 0.0403 |
> | BCW | Acc | 0.9912 | 0.9474 | 0.9474 |
>
> Table 2: Impact of Different Levels of Regularization on Model Accuracy
>
> **W3 & Q1:** Complexity
>
> **A3:** On conventional datasets, the complexity of KANFIS is comparable to that of ANFIS, with training efficiency significantly surpassing that of KAN. However, as the dimensionality and size of the dataset increase, the traditional ANFIS model begins to suffer from dimensionality explosion, and the training difficulty of KAN also increases, ultimately rendering both ANFIS and KAN impractical for high-dimensional, large-scale datasets. In contrast, KANFIS demonstrates a clear advantage. We have included a comparison of training time vs. accuracy and parameter count vs. FLOPs, which are presented in Tables 3 **( Time, Acc )** and 4 **( Para, FLOPs )** , respectively.
>
> | Dataset | IT2-KANFIS | T1-KANFIS | MLP | ANFIS | KAN |
> |-|-|-|-|-|-|
> | CCPP | 0.3914, 4.124    | 0.356, 3.9542   | 0.0902, 4.1883 | 0.3042, 3.998 | 4.8063, 3.9313 |
> | BCW | 0.0212, 0.9912   | 0.0207, 0.9912  | 0.0067, 0.9912 | - | 1.4943, 0.9736 |
> | Spam | 0.1357, 0.9392   | 0.1175, 0.9381  | 0.0475, 0.9175 | - | -  |
>
> Table 3: Time-Accuracy Comparison of Models. The training time is measured in seconds per training epoch. For regression datasets, accuracy is measured using RMSE, while for classification datasets, accuracy is measured using Acc.
>
> | Dataset | IT2-KANFIS | T1-KANFIS | MLP | ANFIS | KAN |
> |-|-|-|-|-|-|
> | CCPP | 521, 2600 | 361, 1640 | 193, 480   | 429, 1274    | 1600, 9972 |
> | BCW | 4822, 24330 | 3986, 18396 | 1058, 2208 | 5139, 15398  | 2240, 11518 |
> | Spam | 7314, 36960 | 5034, 23280 | 1922, 3936 | -  | - |
>
> Table 4: Comparison of Parameter Count and FLOPs for Different Models.
>
> **W4:** Comparison with Baselines
>
> **A4:** In the paper, we discuss the comparison between KANFIS, KAN, and ANFIS, and present favorable experimental results. The reason for not including comparisons with models such as GAMI-Net is that we consider KANFIS and GAMI-Net to be fundamentally different types of explainable models. The interpretability of KANFIS arises from its ability to consistently produce IF-THEN rules. In response to your suggestion to include additional baseline comparisons, we have added a comparison with these models, which is presented in Table 5.
>
> | Dataset   | |Ours| NAMs   | GAMI-NET | NODE-GAM |
> |-|-|-|-|-|-|
> | Parkinsons | RMSE | **0.0397** |0.0408 | 0.0403 | |
> | BCW | Acc | **0.9912**|0.9651 | 0.9649 | 0.9737 |
>
> Table 5: Comparison with Additional White-Box Models. Due to incomplete open-source availability and time constraints, some models do not perform well on certain datasets.

---

> > ### Author Rebuttal · Reviewer_Q6nn · 2026-04-07
> >
> > Thank you for the detailed rebuttal. I have read the authors’ response carefully, and my main concerns have been adequately addressed. The clarifications and additional evidence strengthen the paper and improve my confidence in the technical soundness of the work. Based on the rebuttal, I maintain my positive assessment of the paper.

---

> > > ### Author Response · Authors · 2026-04-07
> > >
> > > We sincerely thank you for your recognition and valuable guidance. We will supplement the datasets you suggested, along with additional baseline comparisons and complexity analyses, either in the main text or in the appendix. With your support, we are confident that we can further improve the manuscript in the final version.

---

### Decision · Program_Chairs · 2026-04-30

**Decision:**

Accept (regular)

**Comment:**

This paper proposes KANFIS, a neuro-symbolic architecture that integrates the additive decomposition of Kolmogorov-Arnold Networks (KAN) into Adaptive Neuro-Fuzzy Inference Systems (ANFIS). By shifting from product-based to additive aggregation, KANFIS successfully breaks the "exponential rule explosion" in high-dimensional spaces, reducing rule complexity to a linear scale.

During the rebuttal phase, the authors actively addressed most reviewer doubts. Initially, reviewers raised concerns regarding the limited scope of the evaluated datasets, the lack of comparisons with recent interpretable baselines, and the potential computational overhead of the proposed architecture. The authors resolved them by supplementing multiple high-dimensional datasets, introducing strong baselines like NAMs and GAMI-Net, and providing quantitative analyses of FLOPs and training time that demonstrate KANFIS's structural efficiency in practical applications.

Some concerns still remain. Reviewers ddT9, and UTbc find the number of datasets limited. However, I believe that such a scope is understandable given the novel architectural shift and the high cost of validating rule transparency in this domain. Another remaining issue from Reviewer UTbc is the lack of dedicated quantitative evaluations for "uncertainty awareness" (e.g., out-of-distribution robustness).

To broaden the discussion, the authors are encouraged to discuss how their continuous fuzzy rule extraction complements consistency-driven paradigms like abductive learning [1,2], thereby offering greater benefits to the integration of neuro-symbolic systems.

[1] Dai et al., Bridging machine learning and logical reasoning by abductive learning, 2019.

[2] Hu et al., Efficient Rectification of Neuro-Symbolic Reasoning Inconsistencies by Abductive Reflection, 2025.